# Synergistic Effects of Microbial Biostimulants and Calcium in Alleviating Drought Stress in Oilseed Rape

**DOI:** 10.3390/microorganisms13030530

**Published:** 2025-02-27

**Authors:** Virgilija Gavelienė, Rima Mockevičiūtė, Elžbieta Jankovska-Bortkevič, Vaidevutis Šveikauskas, Mariam Zareyan, Tautvydas Žalnierius, Jurga Jankauskienė, Sigita Jurkonienė

**Affiliations:** Laboratory of Plant Physiology, Nature Research Centre, Akademijos Str. 2, 08412 Vilnius, Lithuania; rima.mockeviciute@gamtc.lt (R.M.); elzbieta.jankovska@gamtc.lt (E.J.-B.); vaidevutis.sveikauskas@gamtc.lt (V.Š.); mariam.zareyan@gamtc.lt (M.Z.); tautvydas.zalnierius@gamtc.lt (T.Ž.); jurga.jankauskiene@gamtc.lt (J.J.)

**Keywords:** *Brassica napus*, calcium carbonate, commercial plant probiotics, drought stress responses

## Abstract

The study aimed to examine the changes in winter oilseed rape (*Brassica napus* L.) under simulated prolonged drought and to assess the effects of a microbial biostimulant ProbioHumus and calcium, individually and in combination, in order to improve the plant’s drought resistance and to identify the biochemical processes occurring in the plant tissues. The oilseed rape cv. ‘Visby’ was grown under controlled laboratory conditions. CaCO_3_ (hereafter, Ca) (3.71 g) was added to the soil of one pot at 70 g m^−2^. Seedlings at the 3–4 leaf stage were sprayed with ProbioHumus 2 mL 100 mL^−1^ and exposed to drought for 8 days to achieve a high water deficit. Irrigation was then resumed, and recovery was assessed after 4 days. The data showed that the microbial biostimulant alleviated the physiological and biochemical response of oilseed rape to drought stress. ProbioHumus + Ca reduced plant wilting by increasing leaf relative water content (RWC) by 87% and induced drought tolerance by increasing endogenous proline content 4-fold, increasing photosynthetic pigment content in leaves by 10–28%, reducing H_2_O_2_ by 53% and malondialdehyde (MDA) by 45%, and stimulating stomata opening (by 2-fold on the upper and 1.4-fold in the lower leaf surface), vs. drought control. The most effective measure to increase plant survival and/or resume growth after drought was the application of a microbial biostimulant with additional calcium to the soil. The practical implications of this research point to the potential benefits of applying these ecological measures under field conditions.

## 1. Introduction

Microorganisms play an important role in adapting and/or withstanding environmental stresses. Efficient commercial microbial inoculants (probiotics) determine the successful implementation of these beneficial microbes in agriculture, especially when microbes are alive in the formulation products [1,2,3]. Oilseed rape (*Brassica napus* L.) is an economically crucial agricultural crop widely grown in many countries. However, adverse environmental conditions such as drought can limit the growth, development, and survival of oilseed rape plants, especially in regions with low rainfall [4,5]. Drought induces morphological, physiological, and biochemical changes in crops and results in reduced yield, which is a major task that needs to be addressed by finding ecological tools to solve this problem [6]. Currently, strategies to enhance the plant′s ability to cope with drought stress include traditional breeding, the genetic engineering of drought-tolerant transgenic plants, and water-saving irrigation [7,8]. Unfortunately, these strategies are very complex, time-consuming, and labor-intensive. There is a need to focus on organic and eco-friendly methods for improving crops facing drought stress. Many reports reveal that the use of plant probiotic microorganisms, also known as bioprotectants, biofertilizers, or biostimulants, have been extensively associated with stress responses in plants and are important in the control of plant abiotic stress [9,10]. According to the Higa and Parr concept [11], environmentally friendly effective microorganisms (EMs) are beneficial to agricultural fields when used as fertilizer or sprays. EMs consist of carefully selected microorganism species, including dominant populations of lactic acid bacteria and yeasts, along with smaller amounts of photosynthetic bacteria, actinomycetes, and other organisms that can enhance crop growth, yield, and quality [11]. Integrating beneficial microorganisms as biostimulants in sustainable agriculture is, therefore, attracting growing interest. So, one of the questions of this study was to evaluate whether foliar application of the commercial plant probiotic ProbioHumus produced by Latvian consumers improves the response of oilseed rape to prolonged water deficit stress. Probiotics can be applied directly to the soil, via seed treatment or foliar spraying. In addition, calcium (Ca^2+^) is known as the main signal transmitter in controlling plant development and the response to drought stress, especially when Ca levels are low [12,13,14]. Furthermore, its multifaceted roles in the response to drought stress include reducing the degree of lipid peroxidation and increasing the activity of antioxidant enzymes [14,15,16]. Several studies have demonstrated the beneficial effects of exogenous calcium supplementation on stress tolerance [14,15]. This can be related to the role of Ca in water status regulation, stomatal opening and closing, antioxidant activity, osmolyte accumulation, and improvement of photosynthetic pigment contents [17,18,19,20,21,22]. These findings offer new insights into the mechanisms through which exogenous Ca^2+^ improves crop phenotypes under water-limited conditions. However, it is important to note that in calcareous soils, the addition of Ca should be avoided. The combined use of calcium and probiotics is likely to have a synergistic effect and could be a promising approach to improve sustainable crop production and abiotic stress tolerance. The use of such substances to regulate drought tolerance in oilseed rape (*Brassica napus* L.) has not been investigated. We hypothesized that oilseed rape treatment with microbial biostimulants coupled with Ca can benefit growth by strengthening (intensifying) the tolerance mechanisms protecting them from the detrimental effects of drought stress by up-regulating the antioxidant system, osmoregulation, and secondary metabolite accumulation, and increasing the survival.

## 2. Materials and Methods

### 2.1. Plant Growth Conditions and Treatments

Winter oilseed rape (*Brassica napus* L.) cv. ‘Visby’ was grown in controlled laboratory conditions in a Climacell plant growth chamber (Medcenter Einrichtungen GmbH, Planegg, Germany) at 20–23 °C. The plants were illuminated with 60 µmol m^−2^ s^−1^ of cool white fluorescent light at the soil level, with a photoperiod of 16 h light and 8 h dark, and maintained at 65% humidity. The 60 seeds were sown in each pot (40 × 17 × 13) cm containing 6 L high-quality pH adjusted moss peat substrate (pH 5.5–6.5) (SuliFlor SF2, Radviliškis, Lithuania). According to the experimental design, seeds from four pots were primed with a commercial plant probiotic, ProbioHumus (purchased from Baltic Probiotics, Rucava, Latvia) at 2 µL g^−1^ (Figure 1). Peat moisture was monitored using a soil moisture meter (Biogrod, Shanghai, China) and maintained at 70% through daily irrigation. Calcium carbonate CaCO_3_ (MKDS, Vilnius, Lithuania) (hereafter, ‘Ca’) was added to the soil according to the scheme, at a rate of 3.71 g per pot (equivalent to 70 g m^−2^). After 21 days of cultivation, seedlings were sprayed according to the experimental design at the 3–4 leaf stage [23] with ProbioHumus at 2 mL 100 mL^−1^, containing *Bacillus subtilis* (103 CFU/mL), *Saccharomyces cerevisiae*, *Bifidobacterium animalis*, *Bifidobacterium bifidum*, *Bifidobacterium longum* (104 CFU/mL), *Lactobacillus diacetylactis*, *Lactobacillus casei*, *Lactobacillus delbrueckii* (105 CFU/mL), *Lactococcus lactis* (102 CFU/mL), *Streptococcus thermophilus*, *Rhodopseudomonas palustris*, and *Rhodopseudomonas sphaeroides* (104 CFU/mL). For the drought stress control studies, plants were subjected to an 8-day prolonged drought period to induce a significant water deficit. Afterward, irrigation was resumed to bring soil moisture back to 70%, and plant recovery was subsequently assessed.

### 2.2. Experimental Design

Eight pots were used for the experiment: four for watering and four for simulating prolonged drought (with two replicates). The experiment was conducted three times. Calcium was applied to the soil at a rate of 3.71 g per pot, equivalent to 70 g m^−2^. ProbioHumus was used for seed priming and seedling spraying at 2 mL per pot, following this scheme: (1) watered control, (2) ProbioHumus and watering, (3) Ca and watering, (4) Ca + ProbioHumus and watering, (5) drought control, (6) ProbioHumus and drought, (7) Ca and drought, and (8) ProbioHumus + Ca and drought.

### 2.3. Sampling

Rapeseed seedlings were collected on three occasions: on the 4th day of drought (40% peat moisture), on the 8th day of prolonged drought (20% peat moisture), and on the 12th day of the experiment, after irrigation was resumed on the 8th day (70% peat moisture). The watered plants, used as controls, were sampled at the same time (pea moisture 70%). For biochemical analysis, three independent replicates were conducted using the third leaves of oilseed rape seedlings. Freshly harvested oilseed rape leaves were used for ethylene emission analysis and pigment measurement. For plasma membrane ATPase (PM ATPase) activity, MDA, H_2_O_2_, and proline assays, the samples were immediately frozen in liquid nitrogen and stored in a low-temperature freezer (Skadi Green line) at −80 °C until the analysis.

### 2.4. RWC Measurement

The plant water status was assessed by measuring the leaf RWC of control and drought-stressed plants, as described in [24]. The first measurement was taken immediately after collecting the plant material to determine the fresh weight (W_f_). The second measurement was taken after soaking the plant material in distilled water for 24 h to obtain saturated weight (W_t_). The dry weight (W_d_) was determined by drying the leaves in a drying chamber at 70 °C, followed by weighing. The RWC was calculated using thirty plant leaves per treatment from three biological experiments, based on the following equation:RWC = [(W_f_ − W_d_)/(W_t_ − W_d_)] × 100%(1)

### 2.5. Evaluation of Stomatal Density and Opening

Stomatal measurements were performed on fully expanded leaves using the leaf impression technique, as described by Hilu and Randall [25]. A layer of acrylic (synthetic nail coating) was applied to both the adaxial and abaxial leaf surfaces, allowed to dry briefly, then removed carefully and mounted for microscopy. The stomatal density (number in mm^2^) and percentage of closure stomata were examined under a microscope (magnification ×20 and ×40) (Olympus BX51, Olympus, Tokyo, Japan) from three microscope fields randomly selected for each leaf epidermis. Three microscope fields were randomly selected for each leaf epidermis, and three measurements were taken from random spots within each print. The average of these three readings was used to calculate stomatal density for each leaf.

### 2.6. Estimation of Photosynthetic Pigment Content

Photosynthetic pigments (Chl a, Chl b, total Chl, and carotenoids) were extracted with N, N’-dimethyl-formamide (DMF) (Sigma-Aldrich, St. Louis, MO, USA) and estimated according the absorbance of pigments measured at 480, 664, and 647 nm [26].

### 2.7. Measurement of MDA Content

The Hodges [27] method was used to estimate MDA. Leaf tissue (0.3 g) was homogenised in 5% trichloroacetic acid (TCA) (Sigma-Aldrich, St. Louis, MO, USA). The homogenate was centrifuged at 13,000× *g* for 17 min (MPW-351 R centrifuge), and the supernatant was mixed with 20% TCA containing 0.50% thiobarbituric acid (TBA) (Alfa Aesar, Haverhill, MA, USA). The mixture was incubated in a water bath (Blockthermostat BT 200) at 95 °C for 30 min, then cooled on ice. Absorbance was measured at 532 nm using a spectrophotometer (Analytik Jena Specord 210 Plus, Analytik Jena, Jena, Germany), with corrections for interference at 660 nm. The results were expressed in µmol g^−1^ FW. To account for drying effects, RWC was used as a coefficient to normalize the MDA content, allowing for a more accurate comparison between treatments.

### 2.8. Estimation of Hydrogen Peroxide (H_2_O_2_) Content

The method of Velikova et al. [28] was used to determine the H_2_O_2_ content in leaves. Leaf material (0.3 g) was homogenized in 5% TCA (Sigma-Aldrich, St. Louis, MO, USA) and centrifuged at 10,000× *g* for 5 min. The supernatant was mixed with potassium phosphate buffer (10 mM, pH 7.0) (Alfa Aesar) and potassium iodide (1 M) (Alfa Aesar) in a 1:1:2 ratio. The reaction mixture was incubated for 30 min at 25 °C in the dark. Absorbance was measured at 390 nm using a spectrophotometer. The H_2_O_2_ content was calculated from the standard curve and expressed in µmol g^−1^ FW.

### 2.9. Estimation of Proline Levels

The amount of free proline in frozen rapeseed leaves was determined using ninhydrin [29]. A supernatant (0.5 g) was prepared by mixing equal volumes of plant material, acetic acid, and acidified ninhydrin. This mixture was heated at 108.5 °C, and the resulting chromophore was extracted with toluene. Proline concentrations were determined by spectrophotometric measurement at 520 nm using a multilayer quartz cuvette and a Rainbow microplate reader. Calibration was performed with L-proline (SigmaAldrich Chemie, GmBH, Steinheim, Germany). The results were expressed in µmol g^−1^ FW.

### 2.10. Estimation of Ethylene Levels

Endogenous ethylene emission from fresh leaves was estimated according to Child et al. [30] with minor modifications. Leaf tissues (0.25–0.45 g FW) were placed in 40-mL glass vials (Agilent Technologies, Santa Clara, CA, USA), sealed with PTFE/Si septum caps and incubated for 24 h at 21 °C in the dark. After incubation, a 1 mL gas sample was withdrawn with a gas-tight syringe (Agilent Technologies) and injected into a gas chromatograph equipped with a stainless-steel column (Propac R, Sigma-Aldrich, USA) and a hydrogen flame ionization detector. The injector, column, and detector temperatures were set to 110, 90, and 150 °C, respectively. Helium (AGA) was used as the carrier gas, and a calibration was performed with an ethylene standard (Messer, Bad Soden, Germany). Ethylene emission rates were expressed in nL g^−1^ FW h^−1^.

### 2.11. Estimation of PM ATPase Activity

The membrane-enriched fraction (microsomes) was isolated from oilseed rape leaf samples by differential centrifugation: (1) at 4500× *g* for 5 min, (2) at 18,000× *g* for 20 min (centrifuge MPW-351 R), and (3) supernatant at 92,200× *g* for 1 h (centrifuge Sorvall WX 100 Ultra. Thermo Scientific, Waltham, MA, USA)). Protein content was measured using the Bradford dye binding procedure [31] at 595 nm. The H^+^-ATPase activity of the microsomal fraction was assessed by measuring the release of inorganic phosphate (P_i_) resulting from ATP hydrolysis [32]. The P_i_ reaction was measured using ammonium molybdate and stannous chloride at 750 nm. PM ATPase activity was expressed as μmol P_i_ mg^−1^ protein h^−1^.

### 2.12. Determination of Plant Survival

Plant survival was assessed after a 12-day recovery period, during which plants were considered dead if they did not recover from drought stress. Recovery was confirmed by the generation of new green shoots at the apical meristem of the rosette leaf. Plant survival was expressed as the percentage of plants that recovered, calculated from the total number of plants.

### 2.13. One-Way Statistical Analysis by ANOVA

The results are presented as means of three independent experiments, each with a minimum of two replicates. The data analysis was performed using one-way analysis of variance (ANOVA). The data satisfied the assumptions of normality and homogeneity of variance. Tukey’s test was applied to assess the statistical significance of differences between means (*p* < 0.05).

## 3. Results

### 3.1. Effect of Microbial Biostimulant and Ca on Physiological and Biochemical Indicators of Control and Drought-Stressed Plants

#### 3.1.1. Leaf RWC

Prolonged drought significantly (*p* < 0.05) reduced leaf RWC to 38%. Exogenous application of ProbioHumus, Ca, and ProbioHumus + Ca reduced the effects of 8 days of drought from 38% to 73%, 72%, and 87%, respectively. After 4 days of renewed irrigation, the RWC of drought-exposed ProbioHumus- and Ca-treated plants increased to the RWC of the regularly irrigated plants, except for the drought control, whose RWC did not recover fully (Figure 2a).

#### 3.1.2. Ethylene Level

Prolonged drought promoted ethylene release from oilseed rape leaves compared to the irrigated control plants. After 4 and 8 days of drought, the ethylene content increased by 3.7- and 5.6-fold, respectively. ProbioHumus, Ca, and ProbioHumus + Ca treatments significantly reduced ethylene emission in leaves by 27%, 24%, and 38% (*p* < 0.05) compared to plants exposed to 8 days of drought. Drought exposure was best reduced in ProbioHumus + Ca treated plants, where the ethylene content was 41% lower than in drought-only treated plants. After 4 days of recovery by irrigation, the ethylene content of plants treated with the drought-protective compounds decreased more than that of untreated and drought-stressed plants (Figure 2b). The ethylene emission of continuously irrigated plants was independent of their treatment with drought protectants.

#### 3.1.3. H_2_O_2_ and MDA Content

A prolonged drought of 8 days caused a pronounced increase in the ROS (H_2_O_2_) levels in oilseed rape leaves from 5 to 19 nmol g^−1^ FW (Figure 2c), which led to lipid peroxidation, as assessed by MDA, which increased from 11 to 30 nmol g^−1^ FW (Figure 2d). Drought-protective compounds reduced H_2_O_2_ release almost 2-fold: ProbioHumus + Ca had the greatest effect, reducing H_2_O_2_ by 53%. MDA content was also reduced by ProbioHumus by 25%, Ca by 24%, and ProbioHumus + Ca by 45% compared to the drought control. In addition, H_2_O_2_ levels were reduced 2-fold after 4 days of recovery from irrigation compared to the 8-day drought control. The applied microbial biostimulant ProbioHumus, as well as Ca and ProbioHumus + Ca, further reduced the H_2_O_2_ levels in oilseed rape leaves after 4 days of renewed irrigation, and the MDA level was also reduced to the level of continuously irrigated plants (Figure 2c,d).

#### 3.1.4. Endogenous Proline Content

The endogenous proline content of drought-stressed oilseed rape leaves increased with the duration of drought, reaching 5.26 µmol g^−1^ FW during the first 4 days of drought, and then increasing almost 3-fold during the next 4 days compared to the irrigated control plants (Figure 2e). Both ProbioHumus and Ca, individually and in combination, increased the proline content in oilseed rape leaves in both stages of simulated drought vs. drought control. The highest proline content of 15.26 µmol g^−1^ FW was found in the leaves of plants treated with ProbioHumus + Ca + drought. When irrigation was resumed, the free proline content decreased rapidly, and after four days, was close to that found in unstressed plants.

#### 3.1.5. PM ATPase Activity

Under drought stress, PM ATPase activity in oilseed rape cells decreased by up to 6-fold compared to control plants. Drought-protective compounds at the beginning of the simulated drought (after 4 days) maintained PM ATPase activity similar to that of continuously irrigated plants. In oilseed rape grown with added Ca, both with and without ProbioHumus application, PM ATPase activity decreased almost to that of the drought control after 8 days of simulated drought (Figure 2f). In contrast, plants treated with ProbioHumus recorded higher H^+^-ATPase activity (47.5% and 44.4% on average) during the entire drought period vs. all other test variants. After 4 days of recovery by irrigation, the PM-ATPase activity of drought-treated plants increased 4.5-fold and approached that of continuously irrigated plants, as well as that of plants treated with drought-protective compounds (Figure 2f).

#### 3.1.6. Photosynthetic Pigment Content

A quantitative analysis of leaf pigments in oilseed rape showed a significant decrease in chlorophylls a and b after 4 and 8 days of simulated drought compared to continuously irrigated plants. Foliar application with ProbioHumus, Ca additive in soil, and ProbioHumus with soil Ca mitigated the drought-induced decrease in chlorophyll a and b concentrations, i.e., their concentrations became significantly higher by 10–28% (*p* < 0.05) vs. the drought control. All protectants increased carotenoid content after 8 days of drought: Carotenoid concentrations increased by 0.09 mg g^−1^ FW, 0.10 mg g^−1^ FW, and 0.11 mg g^−1^ FW in plants treated with ProbioHumus, Ca, and ProbioHumus + Ca, respectively. Pigment content increased significantly with renewed irrigation, especially in plants treated with anti-drought treatments (Table 1). The pigment content of continuously irrigated plants was independent of the treatment with drought-protecting compounds.

### 3.2. Effect of Microbial Biostimulant and Ca on Stomata Density and Aperture in Leaves of Control and Drought-Stressed Plants

Measurements of the change in stomatal density and aperture allow us to characterize the response of the plant to various environmental stimuli. In irrigated plants treated with ProbioHumus in combination with Ca, the density of stomata on the upper and lower leaf surface was higher than in the control samples. After 4 days of drought due to wilting, the number of stomata on the upper and lower leaf surface was reliably higher than in the irrigated control plants. In the upper side of oilseed rape leaves treated with ProbioHumus and Ca + drought, the density of stomata varied during the prolonged drought: After the first 4 days, it remained very similar to the irrigated control, and after 8 days of drought, it increased almost to that of the drought control on day 4 of drought (Figure 3a,b). Microbial biostimulants stimulated opening the stomata of the irrigated control plants: The percent of open stomata both in the upper and lower surface of control leaves was less vs. plants affected by biostimulants (Figure 3c,d). After 4 days of drought, in the upper and lower surface of plants treated with ProbioHumus in complex with Ca, the number of open stomata was 30% and 50% higher, respectively, vs. drought control. After 8 days of drought stress, the percent of open stomata was the lowest in the lower surface of the leaf; still, microbial biostimulants increased the stomata opening to that of the irrigated control without treatment. In recovered plants, the percent of stomatal opening also recovered to that of the irrigated control at the beginning of the experiment, except for the ProbioHumus + Ca test variant, where the stomata opening increased about 1.5–2 fold vs. control plants (Figure 3c,d).

### 3.3. Effect of Microbial Biostimulant and Ca on Survival of Control and Drought-Stressed Plants

After 8 days of simulated drought stress and 12 days of irrigation recovery, the number of surviving oilseed rape plants treated with ProbioHumus and Ca was significantly higher (on average 42%) compared to plants treated with drought stress alone. The number of plants treated with ProbioHumus + drought was lower than those treated with Ca + drought, and these differences were statistically significant (Table 2).

## 4. Discussion

In regions with low rainfall, droughts limit the growth, development, and survival of oilseed rape [33,34]. Prolonged drought stress adversely affects plant morphological, physiological, biochemical, and molecular processes, leading to growth inhibition. Biologically active compounds can activate metabolic pathways, enhance plant adaptation to drought, and overcome or delay the most important negative effects [3,35,36]. Plant-derived microbial stimulants have been shown to have a high potential to induce plant tolerance to various abiotic stresses and, thus, improve plant growth and productivity-related parameters [3,37]. Furthermore, literature data suggest that exogenous application of calcium has a positive effect on plants growing under drought stress conditions, promoting growth and antioxidant activity [38,39]. The idea of this study was to investigate the effect of the plant biostimulant ProbioHumus, Ca, and their combination on growth, biochemical responses, and recovery of oilseed rape after simulated prolonged drought under laboratory conditions.

Various cytological, physiological, and biochemical indicators, such as stomatal density and openness, photosynthetic pigment levels, enzyme activity, concentrations of primary and secondary metabolites, and overall plant growth can be used to assess plant responses to environmental factors, even when these responses are not externally visible. According to the literature, the RWC of leaves is considered an indicator of the plant’s water status and dehydration [40,41]. RWC has been reported to decrease by more than 40% in various plants under stress conditions [42,43]. In our study, 8 days of prolonged drought reduced the RWC of oilseed rape leaves by up to 38%. Several studies suggest that plant microbial biostimulants and exogenous Ca^2+^ treatments can help increase the RWC of crops and, to some extent, compensate for drought-induced water deficit [44,45]. Our findings support this, as treatments with ProbioHumus and ProbioHumus + Ca mitigated drought-induced wilting, with RWC doubling in these plants. A reduction in the RWC causes stomatal closure, which, in turn, limits photosynthesis. Drought stress alters stomata density across different plant species, although the data are sometimes contradictory [46]. Some literature suggests that drought stress may reduce stomata density, as seen in *Hordeum vulgare*, where reduced stomatal density may enhance water stress tolerance [47]. This is not always the case, and other research indicates that a higher density of smaller stomata can reduce water loss and improve drought tolerance in crops [48,49]. So, drought can both increase stomatal density and reduce stomatal opening [50]. We observed a significant increase in the number of stomata in oilseed rape leaves during the drought. Stomatal density was highest on the lower-leaf surface of plants treated with ProbioHumus + Ca + drought. These results align with findings from studies on olive leaves, where microbial inoculation led to increased stomatal density [51]. In recovered plants treated with ProbioHumus + Ca, the stomatal opening was about 2-fold higher than in continuously watered control plants.

Drought has been shown to cause a reduction in chlorophyll content in the leaves of various crops such as rice and oats [52,53]. Reduced chlorophyll content due to drought stress can lead to the inactivation of photosynthesis. Microbial plant biostimulants and exogenous calcium have been shown to modify pigment accumulation in plant leaves and play a key role in tolerating/resisting drought stress [54]. Our data show that the pigment content in oilseed rape leaves was significantly reduced after 4 and 8 days of prolonged drought compared to irrigated control. The foliar application of ProbioHumus, Ca in the soil, and ProbioHumus + Ca mitigated the drought-induced decrease in chlorophyll a and chlorophyll b concentrations. Pigment content increased significantly with renewed irrigation, especially in plants treated with drought-protective compounds. The decrease in chlorophyll content due to drought stress is due to photo-oxidation of the pigment and chlorophyll degradation, which is considered a sign of oxidative stress [5].

Thus, when plants are exposed to stressful environmental conditions, ROS production increases and can cause severe cell damage. During drought stress, H_2_O_2_ is overproduced among all ROS [55]. In this study, a simulated prolonged drought caused a significant increase in H_2_O_2_ content in oilseed rape leaves from 5 to 19 nmol g^−1^ FW, resulting in an increase in lipid peroxidation, as assessed by MDA, from 11 to 30 nmol g^−1^ FW. On the other hand, it has been observed that low concentrations of H_2_O_2_ can be rapidly formed in plant cells in the presence of exogenous Ca^2+^ [16], then H_2_O_2_ molecules generate signals that trigger a series of protective physiological responses to reduce drought stress. In this study, all the drought-protecting compounds reduced H_2_O_2_ release by almost 2-fold: ProbioHumus + Ca had the highest effect, reducing H_2_O_2_ by 53% and MDA by 45% compared to the drought control. The microbial biostimulants, as well as Ca alone and in complex, reduced H_2_O_2_ and content in oilseed rape leaves after 4 days of renewed irrigation to the level of continuously irrigated plants. Similar data on probiotic effects on plants have been obtained in studies with other plants [56,57].

Several publications indicate that an increase of ethylene production after drought stress can reduce photosynthesis, alter growth, inhibit shoot and leaf development, and regulate ROS scavenging by modulating the synthesis of non-enzymatic and enzymatic antioxidants [58,59]. However, it has also been reported that low concentrations of ethylene facilitate the activation of defense signals in plants, while high concentrations inhibit defense signals in wheat [60]. We found that under drought stress, ethylene content in oilseed rape leaves increased intensively to 84.7 nL g^−1^ h^−1^, while in irrigated control plants, it was only 22.7 nL g^−1^ h^−1^. Drought-protective compounds significantly reduced ethylene emission under prolonged drought. Drought exposure was best reduced in ProbioHumus + Ca-treated plants, where the ethylene content was 41% lower than in drought-only treated plants, and at these concentrations, ethylene can act by activating defense signals. When irrigation was resumed, the ethylene emissions of drought-stressed plants were very similar to those of continuously irrigated plants. Plants treated with drought-protective compounds showed a greater decrease in ethylene emissions than untreated and drought-stressed plants. Our data contribute to the findings of the researchers, who suggest that the inoculation of probiotic microorganisms affecting ethylene content may help to remove the inhibitory effect of drought stress on plant growth [61]. These findings were also consistent with the results from [58,62] suggesting that changes in ethylene content during drought can activate the plant’s antioxidant defense system, leading to a reduction in oxidative stress and a concomitant recovery of the plant’s photosynthetic efficiency.

Drought stress usually correlates with oxidative and osmotic stresses, leading to ion imbalance that further leads to drastic changes in cell membrane structure and various other cellular functions [63]. We found data that PM ATPase is an important component of membrane integrity in response to abiotic stress [64,65], and that H^+^-ATPase activity can increase or decrease when plants are exposed to drought stress [66]. Also, several reports described that the treatment of leaves with stress-protective compounds such as polyamines significantly increased PM ATPase activity compared to untreated plants under drought stress [67]. The results of our investigations showed that PM H^+^-ATPase activity in oilseed rape plants under drought stress significantly decreased, by as much as 304.66%, compared to irrigated plants. In plants grown in soil with incorporated Ca, both with and without ProbioHumus, PM ATPase activity decreased in the later stage of drought. Meanwhile, higher H^+^-ATPase activity was recorded during the entire drought period (47.5% and 44.4% on average) in the cells of the plants that were exposed to ProbioHumus than the corresponding variant plants that were watered.

Moreover, the production and accumulation of proline is an important component of the drought resistance mechanism [68,69]. The results of our study showed that proline levels were significantly higher in plants subjected to drought stress than in control plants. In the leaves of oilseed rape plants under drought stress, the proline content increased as the drought prolonged: It reached 5.26 µmol g^−1^ FW during the first 4 days of drought and increased to 10.56 µmol g^−1^. The application of probiotic ProbioHumus and calcium increased proline content in drought-stressed plants. In addition, during a period of longer drought, the synergistic effect of ProbioHumus + Ca begins to emerge, when a slightly higher amount of proline is detected compared to the application of either ProbioHumus or Ca alone. This suggests that increased proline accumulation in rapeseed exposed to ProbioHumus + Ca under drought may improve the efficiency of plant osmotic regulation, which could explain the higher RWC of plants. Several mechanisms are known by which plant probiotics induce plant tolerance to drought stress through osmoprotectants: Some of them produce osmolytes acting synergistically with plant, while others induce synthesis of osmoprotectants in host plants [70]. Our results are also consistent with the authors’ data showing that Ca addition to drought-stressed plants increases proline accumulation [38,71,72] and reinforces the conclusion of [38] that calcium may be involved in osmotic adjustment by increasing proline content in plants under drought stress. While accumulating evidence suggests that both plant probiotics and calcium can increase plant tolerance to stress [73], the precise mechanism behind their combined effect on abiotic stress tolerance remains unclear. A comprehensive understanding of the molecular basis for this synergistic relationship is still lacking. The results of this study showed that both probiotics and Ca increased the survival of oilseed rape after prolonged drought exposure followed by re-irrigation, with the highest survival rates observed when probiotics when probiotics were combined with Ca. These findings underscore that the application of microbial biostimulants and calcium induces biochemical, physiological, and morphological changes in plant tissues, contributing to the enhanced survival of rape. The practical implications of this research point to the potential benefits of applying these ecological measures under field conditions.

## 5. Conclusions

Microbial biostimulant ProbioHumus, both individually and in combination with Ca, enhanced the tolerance mechanisms of oilseed rape (*Brassica napus* L.) against the damaging effects of drought stress. ProbioHumus and ProbioHumus + Ca controlled the drought tolerance of oilseed rape by suppressing drought-induced wilting, regulating stomatal opening, the antioxidant system, osmoregulation, and the accumulation of secondary metabolites, which ensure the survival of plants. Drought stress was best reduced in ProbioHumus + Ca-treated plants, leading to a significant reduction in oxidative stress and a concomitant increase in survival.

## Figures and Tables

**Figure 1 microorganisms-13-00530-f001:**
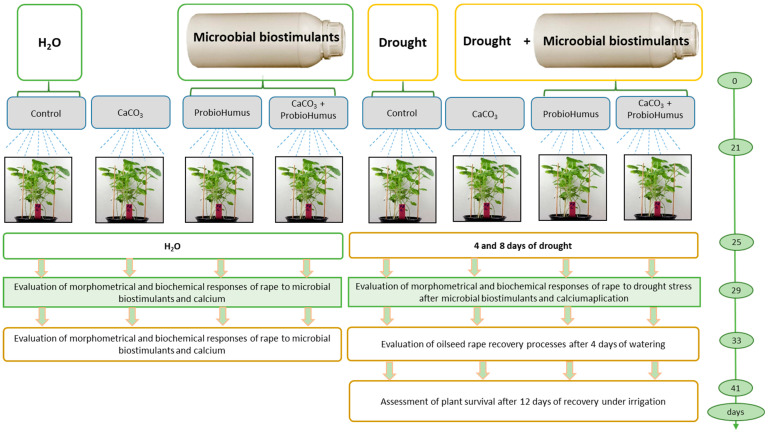
Experimental design. Sixty plants per pot were used, each variant was replicated twice, and three biological experiments were performed.

**Figure 2 microorganisms-13-00530-f002:**
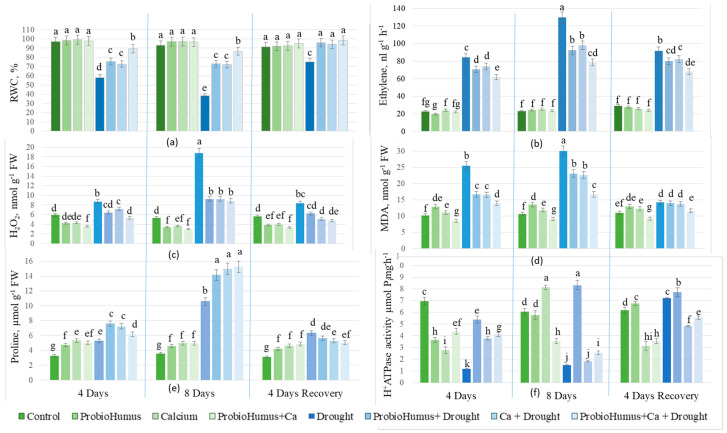
Effect of microbial biostimulant and soil calcium on RWC (**a**), ethylene emission (**b**), H_2_O_2_ (**c**), MDA (**d**), proline (**e**) accumulation, and PM ATPase activity (**f**) in leaves of *Brassica napus* plants after 8 days of prolonged drought and after recovery from 4 days of watering. The values presented are the mean and standard deviation of thirty plant leaves (*n* = 30) from three biological experiments. Means marked with different letters on the same day of drought are significantly different from the watered untreated control (*p* < 0.05).

**Figure 3 microorganisms-13-00530-f003:**
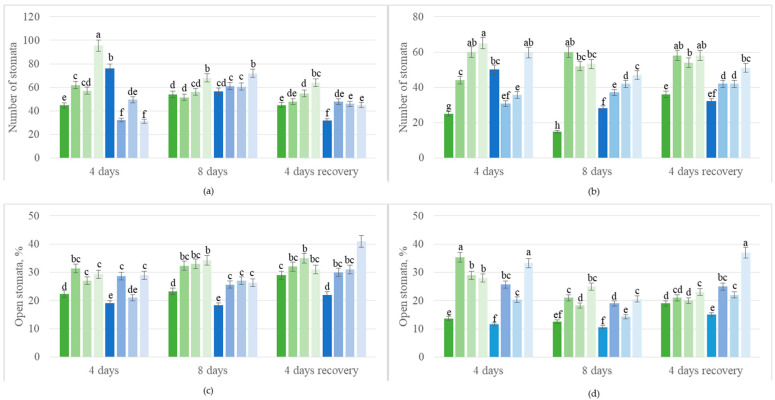
Effect of microbial biostimulant and soil calcium on stomata density (**a**,**b**) and apperture (**c**,**d**) in leaves of *Brassica napus* plants after 8 days of prolonged drought and after recovery from 4 days of watering. The values presented are the mean and standard deviation of thirty plant leaves (n = 30) from three biological experiments. Means marked with different letters on the same day of drought are significantly different (*p* < 0.05).

**Table 1 microorganisms-13-00530-t001:** Effect of microbial biostimulant and soil calcium on chlorophyll and carotenoid contents in leaves of *Brassica napus* after 8 days of prolonged drought and after recovery from 4 days of watering. The results refer to the control group and the drought-stressed group. The values presented are the mean of thirty plants (n = 30) from three biological experiments.

Treatment	Pigment Contents (mg g^−1^ FW)											
	Chlorophyll a			Chlorophyll b			Chlorophyll a/b			Carotenoid		
	4 Days	8 Days	4 Days Recovery	4 Days	8 Days	4 Days Recovery	4 Days	8 Days	4 Days Recovery	4 Days	8 Days	4 Days Recovery
Control	1.42 a	1.41 a	1.43 a	0.37 a	0.38 a	0.38 a	3.84 b	3.71 d	3.76 c	0.36 a	0.36 a	0.37a
ProbioHumus	1.46 a	1.45 a	1.47 a	0.39 a	0.41 a	0.39 a	3.74 c	3.54 e	3.77 c	0.38 a	0.37 a	0.37 a
Calcium	1.44 a	1.45 a	1.46 a	0.38 a	0.38 a	0.37 a	3.79 c	3.82 c	3.95 b	0.38 a	0.37 a	0.37 a
ProbioHumus + Ca	1.43 a	1.42 a	1.47 a	0.39 a	0.40 a	0.41 a	3.67 c	3.55 e	3.59 d	0.39 a	0.36 a	0.38 a
Drought	0.99 d	0.61 c	0.75 d	0.25 c	0.14 c	0.18 d	3.96 a	4.36 a	4.17 a	0.31 b	0.22 c	0.27 b
ProbioHumus + Drought	1.24 b	0.75 b	0.99 c	0.32 b	0.18 b	0.28 b	3.88 b	4.17 b	3.54 d	0.36 a	0.31 b	0.34 a
Ca + Drought	1.09 c	0.72 b	0.83 cd	0.28 c	0.17 b	0.23 c	3.89 b	4.24 b	3.61 d	0.32 ab	0.32 b	0.35 a
ProbioHumus + Ca + Drought	1.23 b	0.75 b	1.11 b	0.32 b	0.19 b	0.31 b	3.84 b	3.95 c	3.58 d	0.37 a	0.34 ab	0.35 a

Means with different letters on the same day of drought are significantly different from the watered untreated control (*p* < 0.05).

**Table 2 microorganisms-13-00530-t002:** Effect of microbial biostimulant and soil calcium on *Brassica napus* plant survival after 8 days of prolonged drought, and after recovery from 12 days of watering. The results refer to the control group and the drought-stressed group. The values presented are the mean of thirty plants (n = 30) from three biological experiments.

Treatment	Number of Survived Plants (%)
Control, H_2_O	100 a
ProbioHumus	100 a
Ca	100 a
ProbioHumus + Ca	100 a
Drought	34 d
ProbioHumus + Drought	60 c
Ca + Drought	69 bc
ProbioHumus + Ca + Drought	76 b

Means with different letters on the same day of drought are significantly different (*p* < 0.05).

## Data Availability

The data supporting the reported results can be found in the archive of scientific reports of the Nature Research Centre https://gamtostyrimai-my.sharepoint.com/:w:/g/personal/sigita_jurkoniene_gamtc_lt/EeNfIKzYBB9GqyYJWA-9cFMBRQvPeA58i_3-gs2SlluWRg?e=8Rfle9, accessed on 24 January 2025.

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
