# Peer review of "Synergistic Effects of Microbial Biostimulants and Calcium in Alleviating Drought Stress in Oilseed Rape"

_microorganisms, 2025, doi:10.3390/microorganisms13030530_

Round 1

Reviewer 1 Report (Previous Reviewer 3)

Comments and Suggestions for Authors

The manuscript addressed application of Microbial Biostimulant and Calcium Combination to Alleviate Drought Stress in Oilseed Rape. The Manuscript is generally good, well-designed and well-written. However the following minor issues need to be resolved before acceptance: 

1- Add a more strong ending sentence in the abstract highlighting the significance of the study.

 2-Instead of saying "significantly reduced," say "reduced by X% (p < 0.05) compared to the control."  Instead of saying "mitigated the drought-induced decrease," specify the amount of mitigation.  Provide actual data values and statistical significance for all key findings.

 3-Use consistent terminology throughout the manuscript.  For example, if you use "relative water content" in one place, don't switch to "RWC" later unless you've clearly defined the abbreviation.  Ensure consistent formatting for units, symbols, and other elements.

4-Enrich the discussion by addressing the potential mechanisms by which the biostimulant and calcium might be exerting their effects.  What are the proposed physiological or molecular pathways involved?  Even if the mechanisms are not fully understood, you can still discuss potential explanations based on existing literature.

5- make sure all the scientific names are written in italic.

Author Response

Dear Prof. Francois Lefort,

With this resubmission, please find our revised manuscript (ID: microorganisms-3490121) for consideration to be published in the Special “Microbial Biostimulants: From the Lab to the Field for a New Agriculture 3.0” of Microorganisms.

We are grateful for the valuable and constructive comments of the reviewers. We have carefully revised the manuscript and made some changes: we have provided point-by-point responses to the comments of reviewers. We rewrote highlighted sentences to avoid overlap. Hopefully, the subsequent modifications are satisfying for all reviewers.

Please find our responses to all the comments and recommendations of reviewers below. We look forward to Your positive response.

Sincerely,

Dr Virgilija Gavelienė and Sigita Jurkonienė

Nature Research Centre, Institute of Botany

Akademijos Str. 2, Vilnius LT-08412, Lithuania

phone: +370 5 2729047

E-mail: virgilija.gaveliene@gmail.com; sigita.jurkoniene@gamtc.lt

Response to Reviewer 1 Comments

Point 1: The manuscript addressed application of Microbial Biostimulant and Calcium Combination to Alleviate Drought Stress in Oilseed Rape. The Manuscript is generally good, well-designed and well-written. However the following minor issues need to be resolved before acceptance:.

Response 1: Thank You for the comment. It is a great pleasure to get such a comment. Following the recommendations, we have edited the text to avoid inaccuracies.

Point 2: Add a more strong ending sentence in the abstract highlighting the significance of the study..

Response 2: Thank You for your comment. We have revised the text according to your recommendation and added the sentence in the end of Abstract.

Point 3: Instead of saying "significantly reduced," say "reduced by X% (p < 0.05) compared to the control."  Instead of saying "mitigated the drought-induced decrease," specify the amount of mitigation.  Provide actual datta values and statistical significance for all key findings.

Response 3: Thank you for your advice. We edited the manuscript according to your recommendation.

Point 4: Use consistent terminology throughout the manuscript.  For example, if you use "relative water content" in one place, don't switch to "RWC" later unless you've clearly defined the abbreviation.  Ensure consistent formatting for units, symbols, and other elements.

Response 4: Thank you for your comment. We edited the text according to your recommendations.

Point 5: Enrich the discussion by addressing the potential mechanisms by which the biostimulant and calcium might be exerting their effects.  What are the proposed physiological or molecular pathways involved?  Even if the mechanisms are not fully understood, you can still discuss potential explanations based on existing literature.

Response 5: Thank you for your valuable comment. We added some thoughts about possible mechanism of plant response to drought, probiotics and Ca.

Point 6: - make sure all the scientific names are written in italic.

Response 6: Thank you for your comment. We checked the scientific names.

Response to Reviewer 2 Comments

Point 1: there are problems with word use    long sentences  etc  so a thorough professional edit would be of value

Response 1: Thank You for valuable comment. The native speaker editor edited the text. We shortened too long sentences.

Point 2: areas of concern are shown with sticky notes

peer-review-44195030.v1.pdf.

Response 2: Thank you for very much for your job. We have edited all the vague expressions that were shown with sticky nots.

Reviewer 2 Report (Previous Reviewer 4)

Comments and Suggestions for Authors

thank you for the now detailed information on replication

there are problems with word use    long sentences  etc  so a thorough professional edit would be of value

areas of concern are shown with sticky notes

Comments on the Quality of English Language

PLEASE on revisions send a clean copy

the english is not optimal

Author Response

Dear Prof. Francois Lefort,

With this resubmission, please find our revised manuscript (ID: microorganisms-3490121) for consideration to be published in the Special “Microbial Biostimulants: From the Lab to the Field for a New Agriculture 3.0” of Microorganisms.

We are grateful for the valuable and constructive comments of the reviewers. We have carefully revised the manuscript and made some changes: we have provided point-by-point responses to the comments of reviewers. We rewrote highlighted sentences to avoid overlap. Hopefully, the subsequent modifications are satisfying for all reviewers.

Please find our responses to all the comments and recommendations of reviewers below. We look forward to Your positive response.

Sincerely,

Dr Virgilija Gavelienė and Sigita Jurkonienė

Nature Research Centre, Institute of Botany

Akademijos Str. 2, Vilnius LT-08412, Lithuania

phone: +370 5 2729047

E-mail: virgilija.gaveliene@gmail.com; sigita.jurkoniene@gamtc.lt

Response to Reviewer 1 Comments

Point 1: The manuscript addressed application of Microbial Biostimulant and Calcium Combination to Alleviate Drought Stress in Oilseed Rape. The Manuscript is generally good, well-designed and well-written. However the following minor issues need to be resolved before acceptance:.

Response 1: Thank You for the comment. It is a great pleasure to get such a comment. Following the recommendations, we have edited the text to avoid inaccuracies.

Point 2: Add a more strong ending sentence in the abstract highlighting the significance of the study..

Response 2: Thank You for your comment. We have revised the text according to your recommendation and added the sentence in the end of Abstract.

Point 3: Instead of saying "significantly reduced," say "reduced by X% (p < 0.05) compared to the control."  Instead of saying "mitigated the drought-induced decrease," specify the amount of mitigation.  Provide actual datta values and statistical significance for all key findings.

Response 3: Thank you for your advice. We edited the manuscript according to your recommendation.

Point 4: Use consistent terminology throughout the manuscript.  For example, if you use "relative water content" in one place, don't switch to "RWC" later unless you've clearly defined the abbreviation.  Ensure consistent formatting for units, symbols, and other elements.

Response 4: Thank you for your comment. We edited the text according to your recommendations.

Point 5: Enrich the discussion by addressing the potential mechanisms by which the biostimulant and calcium might be exerting their effects.  What are the proposed physiological or molecular pathways involved?  Even if the mechanisms are not fully understood, you can still discuss potential explanations based on existing literature.

Response 5: Thank you for your valuable comment. We added some thoughts about possible mechanism of plant response to drought, probiotics and Ca.

Point 6: - make sure all the scientific names are written in italic.

Response 6: Thank you for your comment. We checked the scientific names.

Response to Reviewer 2 Comments

Point 1: there are problems with word use    long sentences  etc  so a thorough professional edit would be of value

Response 1: Thank You for valuable comment. The native speaker editor edited the text. We shortened too long sentences.

Point 2: areas of concern are shown with sticky notes

peer-review-44195030.v1.pdf.

Response 2: Thank you for very much for your job. We have edited all the vague expressions that were shown with sticky nots.

This manuscript is a resubmission of an earlier submission. The following is a list of the peer review reports and author responses from that submission.

Round 1

Reviewer 1 Report

Comments and Suggestions for Authors

Minor Revisions:

Line 28: In my opinion, a more appropriate keyword instead of "prolonged drought simulation" and "stress responses" would be "drought stress" or "drought stress responses".

Line 31: Did the authors mean tolerance or adaptation?

In addition to tolerance, the mechanism of adaptation is more widely known and discussed in the literature than withstanding.

Line 40: To make the sentence easier to interpret, I would advise changing the word "make" to "solve".

Line 47: Spelling error, double comma.

Line 49: Please add a source for the Higa and Parr concept.

Line 62/63: In my opinion, a more appropriate term for H2O2 instead of active oxygen molecules is reactive oxygen species (ROS).

Line 115, 125, 132, 142, 146… and further in the following subsections. Please correct the order of subsections in part of the manuscript: 2. Materials and Methods.

Line 89: Please specify the light of photosynthetically active radiation in the phytotron.
Please provide the lamp data and specify whether it is the unit given for PPF / PPFD or others (the units are different). PPF is measured in micromoles per second, PPFD is measured in micromoles per square meter per second. The device I am using gives PAR in
μmol m-2 s-1.
Please, check and correct.

Line 90: Please provide more details about the peat used for the experiment (manufacturer, composition).

Line 93/94: Please provide the source of the weight method.

Line 94: Please provide the manufacturer CaCO3. What does the abbreviation (MKDS) mean?

Line 93: What type of moisture content is this, I understand that it is by weight, but it should be written down.

Line 199: Please provide the name of the program in which the statistics were calculated.

Line 285: Please complete the description under Table 1 as in Figure 2 (233-235).
Please add in the title of table 1 that the results refer to the control group and the drought-stressed group. The description makes it seem as if the results referred only to the drought-stressed plants.

Please complete the information above and below Table 2 in the same way.

I would also consider changing the names of subsections to emphasize that the Effect of Microbial Biostimulant and Ca was studied in control and stressed plants (3.1, 3.2, 3.3).

Line 323: “The text continues here." should be removed.

Line 346/347: The sentence should be edited, it is incomprehensible. I also do not see the results of the rate of photosynthesis in the work.

Line 411-414: Please cite the source of several reports.

415: Please correct "oilseedrape".

Major Revisions:

In the discussion, it is worth basing it on publications concerning the same species that we are writing about. Differences in species reactions can be significant. There are many publications on drought and biostimulation, and you can certainly find some that refer to rapeseed. You can compare the reaction of rapeseed to olive, rice or oat plants, but you should not base the discussion only on such publications. The manuscript should supplement the literature on Brassica napus plants. Additionally, there is no comparison of the results of specific measurements parameters conducted by the authors of the manuscript to analogous measurements parameters conducted by other scientists.

To sum up, the discussion should be improved.

Author Response

Dear Prof. Francois Lefort,

With this resubmission, please find our revised manuscript (ID: microorganisms-3240477) for consideration to be published in the Special “Microbial Biostimulants: From the Lab to the Field for a New Agriculture 3.0” of Microorganisms.

We are grateful for the valuable and constructive comments of the reviewers. We have carefully revised the manuscript and made some changes: we have provided point-by-point responses to the comments of reviewers. Hopefully, the subsequent modifications are satisfying for all four reviewers.

Please find our responses to all the comments and recommendations of reviewers below. We look forward to Your positive response.

Sincerely,

Drs Virgilija Gavelienė and Sigita Jurkonienė

Nature Research Centre, Institute of Botany

Akademijos Str. 2, Vilnius LT-08412, Lithuania

phone: +370 5 2729047

E-mail: virgilija.gaveliene@gmail.com; sigita.jurkoniene@gamtc.lt

Response to Reviewer 1 Comments

Point 1: Line 28: In my opinion, a more appropriate keyword instead of "prolonged drought simulation" and "stress responses" would be "drought stress" or "drought stress responses".

Response 1: Thank You for the comment. Following the recommendations, we have used the keyword "drought stress responses" instead of "prolonged drought simulation".

Point 2: Line 31: Did the authors mean tolerance or adaptation? In addition to tolerance, the mechanism of adaptation is more widely known and discussed in the literature than withstanding.

Response 2: Thank You for your comment. We have revised the text according to your recommendation.

Point 3: Line 40: To make the sentence easier to interpret, I would advise changing the word "make" to "solve".

Response 3: Thank you for your advice. We changed the word "make" to "solve" as recommended.

Point 4: Line 47: Spelling error, double comma.

Response 4: Thank you for your comment. We edited the text according to your recommendations.

Point 5: Line 49: Please add a source for the Higa and Parr concept.

Response 5: Thank you for your valuable comment. We added a source for the Higa and Parr concept according to your recommendations.

Point 6: Line 62/63: In my opinion, a more appropriate term for H2O2 instead of active oxygen molecules is reactive oxygen species (ROS).

Response 6: Thank you for your comment. We edited the text according to your recommendations.

Point 7: Line 115, 125, 132, 142, 146… and further in the following subsections. Please correct the order of subsections in part of the manuscript: 2. Materials and Methods..

Response 7: Thank you for your valuable comment. We corrected the order of subsections in “Materials and Methods”

Point 8: Line 89: Please specify the light of photosynthetically active radiation in the phytotron.

Response 8: Thank you for your valuable comment. We specified the light of photosynthetically active radiation in the phytotron.

Point 9: Please provide the lamp data and specify whether it is the unit given for PPF / PPFD or others (the units are different). PPF is measured in micromoles per second, PPFD is measured in micromoles per square meter per second. The device I am using gives PAR in μmol m-2 s-1. Please, check and correct.

Response 9: Thank you for the comment. We checked the mistake and corrected to µmol m-2 s-1.

Point 10: Line 90: Please provide more details about the peat used for the experiment (manufacturer, composition).

Response 10: Thank you for your comment. We added manufacturer and composition of the peat used for the experiment.

Point 11: Line 93/94: Please provide the source of the weight method.

Response 11: Thank You for the comment. We corrected the method used.

Point 12: Line 94: Please provide the manufacturer CaCO3. What does the abbreviation (MKDS) mean?

Response 12: Thank You for your comment. We added the name of manufacturer MKDS (Vilnius, Lithuania) of CaCO3 according to your recommendation.

Point 13: Line 93: What type of moisture content is this, I understand that it is by weight, but it should be written down.

Response 13: Thank you for your comment. We changed the sentence “Peat moisture was assessed using a soil moisture meter (Biogrod, China) and maintained at 70%.”

Point 14: Line 199: Please provide the name of the program in which the statistics were calculated.

Response 14: Thank you for your comment. We edited the text according to your recommendations.

Point 15: Line 285: Please complete the description under Table 1 as in Figure 2 (233-235).

Response 15: Thank you for your valuable comment. We completed the description under Table 1 according to your recommendations.

Point 16: Please add in the title of table 1 that the results refer to the control group and the drought-stressed group. The description makes it seem as if the results referred only to the drought-stressed plants. Please complete the information above and below Table 2 in the same way.

Response 16: Thank you for your comment. We edited the text of table caps above and below the tables1 and 2 according to your recommendations.

Point 17: I would also consider changing the names of subsections to emphasize that the Effect of Microbial Biostimulant and Ca was studied in control and stressed plants (3.1, 3.2, 3.3).

Response 17: Thank you for your valuable comment. We corrected the names of subsections according to your recommendations.

Point 18: Line 323: “The text continues here." should be removed.

Response 18: Thank you, we corrected this mistake.

Point 19: Line 346/347: The sentence should be edited, it is incomprehensible. I also do not see the results of the rate of photosynthesis in the work.

Response 19: Thank you for the comment. We edited sentences and corrected mistakes according to your recommendations.

Point 20: Line 411-414: Please cite the source of several reports.

Response 20: Thank you for your comment. We changed cited literature sources.

Point 21: 415: Please correct "oilseedrape".

Response 21: Thank You, it is corrected.

Point 22: In the discussion, it is worth basing it on publications concerning the same species that we are writing about. Differences in species reactions can be significant. There are many publications on drought and biostimulation, and you can certainly find some that refer to rapeseed. You can compare the reaction of rapeseed to olive, rice or oat plants, but you should not base the discussion only on such publications. The manuscript should supplement the literature on Brassica napus plants. Additionally, there is no comparison of the results of specific measurements parameters conducted by the authors of the manuscript to analogous measurements parameters conducted by other scientists. To sum up, the discussion should be improved.

Response 22: Thank you for your valuable comment. We edited the discussion section according to your recommendations.

Response to Reviewer 2 Comments

Point 1: In this study, the authors investigated the potential of using the microbial biostimulant ProbioHumus and calcium to alleviate drought stress in oilseed rape. The research methods are clear, and the results present some meaningful findings, particularly highlighting that the combined application of ProbioHumus and calcium can significantly enhance the recovery ability of oilseed rape after drought. The experimental design is well-structured, utilizing control and treatment groups and measuring various physiological and biochemical parameters.

Response 1: Thank You for your comment. It is a great pleasure to get such a comment.

Point 2: However, there are still some areas in the experimental design and theoretical discussion that need improvement.

Response 2: Thank you for your comment. As recommended, we have edited the description of some areas of the experimental design. We have also improved the discussion.

Point 3: The biggest negative for me is the use of broad statements at key points, to avoid getting into the narrow impact/importance of this work. I think we often want our studies to solve everything, but staying very narrow actually clarifies the importance of a research study, instead of having it feel adjacent to something that would be important to study.

Response 3: Thank you for your valuable comment. We have edited the text to highlight the impact and importance of this work.

Point 4: Line 17: An 8-day drought treatment is considered short-term drought rather than prolonged drought.

Response 4: Thank you for your comment. We edited the text according to your recommendations.

Point 5: Line 200-202: Do the data meet the requirements of normal distribution and homogeneity of variance?

Response 5: Thank you for your valuable comment. We edited the text of “Statistical Analysis” subsection in “Materials and Methods” according to your recommendations.

Point 5: Line 89: Please specify the light of photosynthetically active radiation in the phytotron.

Response 5: Thank you for your valuable comment. We specified the light of photosynthetically active radiation in the phytotron.

Point 6: It is recommended to indicate the number of replicates for each treatment in the figures and tables (n = ?).

Response 6: Thank you for your valuable comment. We indicated the number of replicates for each treatment.

Point 7: Line 205-208: I personally believe that it is not necessary to include this in the results section.

Response 7: Thank you for your valuable comment. We excluded these sentences from the results section.

Point 8: Although the authors measured many drought-related indicators and survival rates, I am curious about the effects of different treatments on plant biomass under drought conditions.

Response 8: Thank you for your comment. We have edited the description of the results to show the relationship between drought-related indicators and survival and photosynthesis and plant growth, which are directly related to biomass accumulation in different treatments.

Point 9: The sample size mentioned in the study is relatively small. Although repeated experiments were conducted, the sample size may not be sufficient to fully reflect the generality of the results. The statistical significance of the results might be limited by the sample size. Additionally, the lack of field application data and the relatively controlled laboratory conditions may limit the study's practical relevance.

Response 9: Thank you for your comment. We have strengthened the statistical description in section MM section according to recommendations. We agree that the practical relevance of the study will be definitively demonstrated by the application of the measures discussed in the paper (probiotics and calcium) in the field under natural conditions. We would like to say that the field trials have already started and that reliable results will be available after at least three years.

Point 10: Although the study demonstrates significant effects of the combined application of ProbioHumus and calcium, the paper does not provide a detailed explanation of the molecular mechanisms behind this synergistic effect. The specific interactions between microbes and calcium that enhance plant drought tolerance still require further investigation.

Response 10: Thank You for the valuable comment. We have strengthened the explanation of the effects of the combined application of ProbioHumus and calcium in a Discussion section. We are now working on specific molecular interactions between microbes and calcium that enhance plant drought tolerance and are planning another article on this.

Point 11: The discussion section still needs revision to provide a deeper interpretation and analysis of the experimental results. Boldly include a discussion of the limitations of the experiment.

Response 11: Thank You for the valuable comment. We have heavily edited the full text of the Discussion.

Response to Reviewer 3 Comments

Point 1: The manuscript addressed the enhancing oilseed rape resilience via microbial biostimulants and calcium in alleviating drought stress. The manuscript is well-designed and well-presented. However, the following points need to be addressed before acceptance: 1) the title should be modified to (Synergistic Effects of Microbial Biostimulants and Calcium in Alleviating Drought Stress in Oilseed Rape)

Response 1: Thank You for Your valuable comment and for such a nice title – now it sounds much better.

Point 2: The motivation of adding calcium should be clearly illustrated along more potential evidence from literature.

Response 2: Thank You for the valuable comment. We edited text according to Your recommendations.

Point 3: Add more details to the Figure 1 legend.

Response 3: Thank You for the valuable comment. We added more information to the Figure 1 legend according to your recommendation.

Point 4: Highlight MORE on how calcium and microbial biostimulants work at both biochemical and physiological levels to enhance drought tolerance in plants.

Response 4: Thank You for the valuable comment. Throughout the text, we have highlighted more biochemical and physiological responses of plants that induce their drought tolerance.

Point 5: Address how Calcium (Ca²⁺) serves as a secondary messenger in response to drought stress, possibly via activating genes that assist in osmotic adjustment and boost antioxidant production, including aquaporins.

Response 5: Thank You for the valuable comment. We highlighted the role of Ca²⁺ as a secondary messenger in response to abiotic stresses according to Your recommendation. We hope to discuss its role in the activation of genes that contribute to osmotic adaptation in next article.

Point 6: Elaborate more on how application of microbial biostimulants like ProbioHumus enhances the expression of antioxidant enzymes, reducing reactive oxygen species (ROS) and oxidative damage. Cite relevant papers approriately.

Response 6: Thank You for the valuable comment. We edited the text about H2O2 and MDA accumulation in oilseed rape leaves.

Point 7: What mechanisms does ProbioHumus use to enhance stomatal sensitivity to environmental cues?

Response 7: Thank You for the comment. We have edited the text of an article related to the effect of ProbioHumus on stomatal sensitivity to the drought.

Point 8: How do soil type and moisture levels influence the effectiveness of these treatments in promoting root growth and nutrient uptake?

Response 8: Thank You for the comment. This study was carried out on a uniform peat substrate, so we cannot draw any conclusions about root growth in different soils.

Response to Reviewer 4 Comments

Point 1: The work showing the importance of Ca amendments is interesting. However, please use a professional science editor as the writing is often clumsy.

Response 1: Thank You for the valuable comment. The manuscript has been consistently edited by a native speaker editor.

Point 2: More details are needed in methods the term soil should not be used.

Response 2: Thank You for Your valuable comment. We have provided more detailed information on the methods used. We have changed the term soil to peat substrate.

Point 3: Your work covers several aspects of known defense measures against stress what is missing and seems vital are effects on growth of the plant and if possible oil content.

Response 3: Thank You for the valuable comment. It’s a pitty, but on this stage of the study, we have no data on oilseed rape oil content.

Point 4: comments made where there are problems or other relevant information are shown as sticky notespeer-review-40517139.v1.pdf

Response 4: Thank You for the valuable comments and work with our manuscript. We carefully revised it.

Reviewer 2 Report

Comments and Suggestions for Authors

General:

In this study, the authors investigated the potential of using the microbial biostimulant ProbioHumus and calcium to alleviate drought stress in oilseed rape. The research methods are clear, and the results present some meaningful findings, particularly highlighting that the combined application of ProbioHumus and calcium can significantly enhance the recovery ability of oilseed rape after drought. The experimental design is well-structured, utilizing control and treatment groups and measuring various physiological and biochemical parameters. However, there are still some areas in the experimental design and theoretical discussion that need improvement.

The biggest negative for me is the use of broad statements at key points, to avoid getting into the narrow impact/importance of this work. I think we often want our studies to solve everything, but staying very narrow actually clarifies the importance of a research study, instead of having it feel adjacent to something that would be important to study. 

Specific:

1. Line 17: An 8-day drought treatment is considered short-term drought rather than prolonged drought.

2. Line 200-202: Do the data meet the requirements of normal distribution and homogeneity of variance?

3. It is recommended to indicate the number of replicates for each treatment in the figures and tables (n = ?).

4. Line 205-208: I personally believe that it is not necessary to include this in the results section.

5. Although the authors measured many drought-related indicators and survival rates, I am curious about the effects of different treatments on plant biomass under drought conditions.

6. The sample size mentioned in the study is relatively small. Although repeated experiments were conducted, the sample size may not be sufficient to fully reflect the generality of the results. The statistical significance of the results might be limited by the sample size. Additionally, the lack of field application data and the relatively controlled laboratory conditions may limit the study's practical relevance.

7. Although the study demonstrates significant effects of the combined application of ProbioHumus and calcium, the paper does not provide a detailed explanation of the molecular mechanisms behind this synergistic effect. The specific interactions between microbes and calcium that enhance plant drought tolerance still require further investigation.

8. The discussion section still needs revision to provide a deeper interpretation and analysis of the experimental results. Boldly include a discussion of the limitations of the experiment.

Comments on the Quality of English Language

I do not comment on the quality of the English, as I trust that the editors will address this issue during the process.

Author Response

(The authors gave the same response as above.)

Reviewer 3 Report

Comments and Suggestions for Authors

The manuscript addressed the enhancing oilseed rape resilience via microbial biostimulants and calcium in alleviating drought stress.

The manuscript is well-designed and well-presented. However the following points need to be addressed before acceptance:

1- the title should be modified to ( Synergistic Effects of Microbial Biostimulants and Calcium in Alleviating Drought Stress in Oilseed Rape)

2- The motivation of adding calcium should be clearly illustrated along the a more potential evidence from  literature.

3- Add more details to the Figure 1 legend.

4- Highlight MORE on how calcium and microbial biostimulants work at both biochemical and physiological levels to enhance drought tolerance in plants.

5- Address how Calcium (Ca²⁺) serves as a secondary messenger in response to drought stress, possibly via activating genes that assist in osmotic adjustment and boost antioxidant production, including aquaporins.

5- Elborate more on how application of microbial biostimulants like ProbioHumus enhances the expression of antioxidant enzymes, reducing reactive oxygen species (ROS) and oxidative damage. Cite relevant papers approriately.

6-What mechanisms does ProbioHumus use to enhance stomatal sensitivity to environmental cues?

7-How do soil type and moisture levels influence the effectiveness of these treatments in promoting root growth and nutrient uptake?

Comments on the Quality of English Language

 Minor editing of English language required.

Author Response

(The authors gave the same response as above.)

Reviewer 4 Report

Comments and Suggestions for Authors

the work showing the importance of Ca amendments  is interesting

However  please use a professional science editor  as the writing is often clumsy 

More details are needed in methods    the term soil should not be used

your work covers several aspects of known defense measures against stress

what is missing and seems vital  are effects on growth of the plant  and if possible oil content 

comments made  where there are problems  or other relevant information are shown as sticky notes

Comments on the Quality of English Language

format  wording  phraseology  and organization  could be greatly improved by thorough editing

Author Response

(The authors gave the same response as above.)
